



# Millennial-age GDGTs in forested mineral soils: [14]C-based evidence for stabilization of microbial necromass

Hannah Gies[1], Frank Hagedorn[2], Maarten Lupker[1], Daniel Montluçon[1], Negar Haghipour[1,3], Tessa Sophia van der Voort[4], and Timothy Ian Eglinton[1]

[1]Department of Earth Sciences, ETH Zürich, Sonneggstrasse 5, 8092 Zürich, Switzerland
[2]Swiss Federal Institute for Forest, Snow and Landscape Research WSL, Zürcherstrasse 111, 8903 Birmensdorf, Switzerland
[3]Laboratory of Ion Beam Physics, ETH Zürich, Otto-Stern-Weg 5, 8093 Zürich, Switzerland
[4]Campus Fryslan, Rijksuniversiteit Groningen, Wirdumerdijk 34, 8911 CE Leeuwarden, Netherlands

**Correspondence:** Hannah Gies (hannah.gies@erdw.ethz.ch)

**Abstract.** Understanding controls on the persistence of soil organic matter (SOM) is essential to constrain its role in the carbon cycle and inform climate-carbon cycle model predictions. Emerging concepts regarding formation and turnover of SOM imply that it is mainly comprised of mineral-stabilized microbial products and residues, however, direct evidence in support of this concept remains limited. Here, we introduce and test a method for isolation of isoprenoid and branched glycerol dialkyl glycerol

tetraethers (GDGTs) – diagnostic membrane lipids of archaea and bacteria, respectively - for subsequent natural abundance radiocarbon analysis. The method is applied to depth profiles from two Swiss pre-alpine forested soils. We find that the $\Delta^{14}C$ values of these microbial markers markedly decrease with increasing soil depth, indicating turnover times of millennia in mineral subsoils. The contrasting metabolisms of the GDGT-producing microorganisms indicates it is unlikely that the low $\Delta^{14}C$ values of these membrane lipids reflect heterotrophic acquisition of [14]C-depleted carbon. We therefore attribute the

[14]C-depleted signatures of GDGTs to their physical protection through association with mineral surfaces. These findings thus provide strong evidence for the presence of stabilized microbial necromass in forested mineral soils.

## 1 Introduction

Soil organic matter (SOM) represents the largest reservoir of carbon in terrestrial ecosystems, exchanging large quantities of

carbon with the atmosphere and supplying aquatic systems with organic and inorganic C (Parry et al., 2007; Battin et al., 2009; Bradford et al., 2016). SOM is comprised of a complex mixture of components that turn over on a wide range of timescales (from seconds to millenia), introducing large uncertainties in climate model predictions (Carvalhais et al., 2014; He and Yu, 2016). Emerging concepts of SOM suggest that only a small fraction of annual C inputs from plants persists in the soils, and that microbial products and residues stabilized by the interaction with reactive minerals comprise the majority of the soil C

pool (Schmidt et al., 2011; Cotrufo et al., 2015; Lehmann and Kleber, 2015; Kallenbach et al., 2016; Kästner and Miltner,





2018). Correspondingly, microbial processes are increasingly being incorporated into soil carbon cycle models (Riley et al., 2014; Ahrens et al., 2015). However, evidence of the 'entombment' of microbial necromass is presently limited and largely circumstantial, being primarily based on the finding of increasing contributions of microbial biomarker as compared plant-derived compounds with increasing soil depth (Amelung et al., 2008; Miltner et al., 2012; Kallenbach et al., 2016; Liang et al., 25 2019; Ma et al., 2018).

Radiocarbon provides valuable constraints on carbon turnover in soils (Trumbore et al., 1996; Schrumpf and Kaiser, 2015), and [14]C measurements are particularly useful when applied at the level of specific compounds (e.g., Van der Voort et al., 2017). Prior radiocarbon analyses of plant-derived biomarkers have indicated their stabilization in mineral soils (Huang et al., 1999; van der Voort et al., 2016), but [14]C-based evidence for stabilization of microbial necromass in SOM is currently lacking. $\Delta^{14}C$ 30 signatures of fatty acids and phospholipid-fatty acids (PLFAs), established indicators for plant and microbial-derived C, suggest active microbial re-synthesis of lipids in deeper soil (Matsumoto et al., 2007; Gleixner, 2013) rather than the stabilization of microbial necromass (Kramer and Gleixner, 2008).

Here we examine the [14]C characteristics of Glycerol Dialkyl Glycerol Tetraethers (GDGTs) – characteristic membrane lipids of microorganisms that are ubiquitous in terrestrial and aqueous environments (Schouten et al., 2013). GDGTs are subdivided 35 into two groups of compounds: isoprenoid GDGTs (isoGDGTs) produced by Archaea (De Rosa and Gambacorta, 1988) and branched GDGTs (brGDGTs) which are of putative bacterial origin (Weijers et al., 2006a) and are especially abundant in soils and peats (Weijers et al., 2006b)(For molecular structures see Figure A1). GDGTs have garnered much attention due to their potential as molecular proxies for environmental conditions: the relative abundance of branched versus isoprenoid GDGTs has been used to qualitatively estimate soil-derived carbon input into marine sediments (Hopmans et al., 2004), while the 40 internal distribution of iso- and branched GDGT isomers carries information of aquatic and soil conditions (Schouten et al., 2002; Powers et al., 2004, 2010; Liu et al., 2013; Coffinet et al., 2014; Yang et al., 2016). For example, the distribution of different brGDGTs, parameterized as the methylation of branched tetraethers (MBT) and cyclisation of branched tetraethers (CBT) indices (Peterse et al., 2012; De Jonge et al., 2014; Naafs et al., 2017), have been found to correlate with mean annual continental air temperature (MAT) and soil pH Weijers et al. (2007), respectively.

Despite their rapid adoption by biogeochemists and paleoclimatologists as valuable molecular tracers and proxies of carbon source and environmental conditions, there are numerous aspects regarding their production, turnover and fate that remain enigmatic. In particular, despite their ubiquity in soils and other environmental matrices, the biological precursors, metabolic processes and physiological drivers giving rise to brGDGT signatures observed in terrestrial and aquatic systems remain poorly constrained. In this context, natural abundance-level radiocarbon measurements of these compounds may provide a valuable 50 approach to better understand their source(s) and turnover rates, while also shedding light on processes that influence their abundance and distribution.

Prior [14]C-based studies of GDGTs have primarily focused on the isoprenoid compounds in marine waters and sediments (Pearson et al., 2001; Smittenberg et al., 2004; Ingalls et al., 2006; Mollenhauer et al., 2007, 2008; Shah et al., 2008). The only reported investigation of branched GDGT [14]C characteristics in lake sediments (Birkholz et al., 2013) yielded $\Delta^{14}C$ values 55 than expected based on the depositional age of the sediment. that were lower than those of the depositional age of the sediment,





although the causes of this pre-aged signal was not established. Although soils are considered a major source of brGDGTs to aquatic systems, the $^{14}$C signatures of brGDGTs in soils has not been determined. Thus, we presently lack crucial information concerning both the production and cycling of this distinctive group of microbial lipids in the context of soil C cycling, as well as the implications for their uses as molecular tracers and proxies.

In the present study, we used molecular-level natural-abundance $^{14}$C measurements to constrain the provenance and turnover of GDGTs in soils. We developed and rigorously tested a preparative high performance liquid chromatography (HPLC) method to isolate both isoprenoid and branched GDGTs from soils (and sediments) for subsequent small-scale radiocarbon analysis by accelerator mass spectrometry (AMS). We then applied this method to samples from two well-studied sub-alpine Swiss soil profiles in order to shed light on their origin and stability. As unequivocal markers of microbial contributions to soils, the
GDGTs provide an opportunity to assess the stability and turnover of microbial biomass in soils.

## 2    Methods

### 2.1    Study Site

Soils were sampled by taking soil cores at two forested sites in Switzerland, one near Lausanne (46.5838°N, 6.6580°E, 800 m asl), and the other one close to Beatenberg (46.7003°N, 7.7623°E, 1490 m asl). Both sites are both part of the Long-term Forest
Ecosystem Research (LWF) network (Innes, 1995) maintained by the Swiss Federal Institute for Forest, Snow and Landscape Research (WSL).

The subalpine soil from the site at Beatenberg is a Podzol which has a thick organic layer followed by a 10 cm A-horizon, and carbonate-free sandstone as parent material. The MAT at the site is 4.6°C and the pH rises from 3.7 to 4.4 with increasing soil depth. The second soil from the Swiss Plateau close to Lausanne is a Cambisol developed on top of a carbonate-containing
moraine. Its A-Horizon extends to 50 cm with a total soil depth of 3 m. Here, the MAT is 7.6°C, the pH is slightly higher compared to the Beatenberg soil and largely invariant (4.6 to 4.5) to a depth of 80 cm (Walthert, 2003).

At each site soil cores were sampled following protocols implemented as part of the LWF sampling program (van der Voort et al., 2016). These soil composites have been previously analyzed for radiocarbon signatures of organic carbon in bulk soil and soil density fractions, as well as in specific alkanes and fatty acids (Van der Voort et al., 2017), which allows the comparison
of the isoprenoid and branched GDGTs with other biomarkers and operationally-defined soil carbon pools.

### 2.2    Reference Materials for Method Validation

Evaluation of the isolation method involved assessment of the purity of separated fractions (i.e., potential interference from other than the desired compounds) and determination of the amount and isotopic composition external contamination introduced in course of the preparation sequence. For the latter, composites of different topsoil samples (0-5 cm) from a 30 x 30
m grassland area in central Switzerland (5.9% C, bulk soil organic carbon (OC) $F^{14}C = 1.155$) and a Rhineland lignite from





the early Miocene (Heumann and Litt, 2002)(62.3% OC, bulk $F^{14}C = 0.003$) were used for assessment and validation with regard to contamination.

## 2.3 GDGT isolation for radiocarbon measurement

Despite the relative ease of detection of GDGTs using modern HPLC-mass spectrometry (MS) techniques, one challenge
in the radiocarbon analysis of GDGTs in soil and sediment samples is their low abundance, with ambient concentrations of brGDGTs and isoGDGTs that are typically in the range of 10 to 1000 ng gdw$^{-1}$ and 1 to 100 ng gdw$^{-1}$ (grams dry weight) soil, respectively (Weijers et al., 2006b). A separation of individual GDGTs of the soil samples used in this study would require on average 3000 g of soil to reach the minimum recommended mass ($\sim 15\ \mu$g C) for high-precision compound-specific radiocarbon analysis (Haghipour et al., 2018). As extracting several kg of material is impractical, focused instead on pooled
isolation and $^{14}$C measurement of isoprenoid GDGTs and branched GDGTs, respectively, at the compound class level, due to the common putative biological percursors and biosynthetic formation pathways for each compound class (Schouten et al., 2013). For this study, the pooling of the GDGTs reduced the required initial sample size to a maximum of 500 gdw of soil. The extraction and purification of the compounds prior to HPLC analysis purification of the compounds prior to HPLC analysis followed a procedure that is similar to that applied to samples processed for quantification of GDGTs (Freymond et al., 2017).
In brief:

lipids were extracted from dried soil samples using a microwave (CEM MARS 5) or an Energized Dispersive Guided Extraction (CEM EDGE) system. No difference in performance was observed for the different extraction systems. Samples were processed in batches of roughly 15 to 20 g of material. For microwave extraction the samples were transferred to the extraction vessels and covered by a dichloromethane (DCM):methanol (MeOH) 9:1 (v/v, 25 ml) solvent mixture. Extraction temperature was
programmed to ramp to 100°C in 35 min and is subsequently held for 20 min. For EDGE extraction 25 ml DCM:MeOH 9:1 (v/v) was used for extraction at 110°C for 2 min and subsequent rinsing with 15 ml followed by a second extraction with 5 ml of solvent at 100°C and rinsing with 35 ml. The process was repeated on additional sample batches to yield sufficient quantities of the target lipid compounds for $^{14}$C analysis. Pooled extracts were then dried under nitrogen flow. After the addition of 5 ml MilliQ water with NaCl the neutral phase was back-extracted with hexane (Hex) and separated on a 1% deactivated silica
column into apolar and polar fractions with Hex:DCM 9:1 (v/v) and DCM:MeOH 1:1 (v/v) respectively. Polar fractions were dried under N$_2$, then re-dissolved in Hex:2-propanol (IPA) 99:1 (v/v) and passed over 0.45 $\mu$m PTFE filters. A portion of the polar fraction was set aside (1%) and an internal $C_4$6 GDGT standard (Huguet et al., 2006) was added to this aliquot to determine GDGT concentrations.

Polar fractions were separated on an Agilent 1260 HPLC-system coupled to an Agilent 1260 fraction collector. Separation was
achieved on two Waters Aquity UHPLC HEB Hilic columns (1.7 $\mu$m, 2.1 x 150 mm) connected in series and preceded by a 2.1 x 5 mm guard column (Hopmans et al., 2016). The columns were heated to 45°C and the flow rate set to 0.2 ml min$^{-1}$. For the first 25 min, compounds elute isocratically with a solvent mixture of 18% Hex:IPA 9:1 (v/v) (solvent A) and 82% hexane (solvent B). For the next 15 min the proportion of solvent B was decreased linearly to 65% followed by a linear gradient to 0% solvent B in 20 min. The total runtime of one injection hence sums up to 60 min followed by 20 min reequilibration





with 82% solvent B. The fraction collection is solely based on retention times with the isoprenoid fraction being collected

from 14.5 to 26 min and the branched fraction from 33 to 43 min (Figure 1). The retention time is recurrently monitored to

avoid undetected drifts. The injection volume is set to 15 $\mu$l corresponding to total GDGT amounts of 100 to 300 ng ml$^{-1}$.

Each sample was injected 10 times and fractions were pooled afterwards. The isolated compound classes and the subset of the

initial polar fraction set aside previously were analyzed for purity and quantification using the same HPLC system coupled

to a quadrupole mass spectrometer (Agilent 6130) according to Hopmans et al. (2016). The isolated fractions were dried and

transferred into 0.025 ml tin capsules (Elementar 03951620). The capsules containing each sample were measured using an

elemental analyzer coupled to a gas-ion-source equipped accelerator mass spectrometer (EA-AMS) (Haghipour et al., 2018) at

the laboratory of Ion Beam Physics at ETH Zürich (Synal et al., 2007; Ruff et al., 2007). In all cases, sample sizes were > 15

$\mu$g C.

### 2.4   Soil turnover model


Turnover times of the individual compounds are calculated based on a steady state two-pool box model (e.g., Trumbore et al.,

1996; Torn et al., 2009; Schrumpf and Kaiser, 2015; van der Voort et al., 2019). This model assumes two homogenous pools

with a first-order decay rate, a fast-cycling and a passive pool. For each of the pools the $F^{14}C$ is calculated independently

(equation 1), where $F^{14}C_{pool(t)}$ is the radiocarbon signal of the respective pool in the sampling year t, lag is the number of

years between $CO_2$ fixation in plants and plant litter entering the soil, $\lambda$ is the radioactive decay of $^{14}C$ (1/8267 years), and

$k_{pool}$ is the decomposition rate constant.

$$F^{14}C_{pool(t)} = F^{14}C_{atm(t-lag)} * k_{pool} + F^{14}C_{pool(t-1)} * (1 - k_{pool} - \lambda) \tag{1}$$

The fraction-weighted sum of the $F^{14}C$ of each of the pools is the modelled $F^{14}C$ of the sample and depends on the

decomposition rate constants of each pool $k_1$ and $k_2$, as well as the relative size of the two pools. The $\Delta^{14}C$ of atmospheric

$CO_2$ was taken from Hua et al. (2013) from 1950 to 1986 and from Hammer and Levin (2017) for the years thereafter.

### 3   Results

### 3.1   Method Validation

Repetitive preparation of samples with 10 injections each reveals a recovery efficiency of 0.85 ± 0.05. Analysis of isolated

fractions on a quadrupole mass spectrometer operated in scan mode (Agilent 6130) for all masses between *m/z* 500 and 1500

reveals that more than 95% of compounds in either fraction are comprised of masses assigned to GDGTs (Figure 1).





The extraneous contamination added in the preparatory process is assumed to be of constant mass $m_c$ and radiocarbon signature $F^{14}C_c$. Therefore, the measured signal $F^{14}C_m$ is a mixture of the sample and the contaminant according to equation 2:

$$F^{14}C_m = \frac{F^{14}C_s * m_s + F^{14}C_c * m_c}{m_s + m_c} \qquad (2)$$

$F^{14}C_s$ and $m_s$ are the true radiocarbon signal and carbon mass of the sample. The measured $F^{14}C_m$ changes depending on the mass of the sample, as smaller masses are more strongly affected by the constant contamination. We assume that in samples with a bulk radiocarbon signal that is either completely modern or does not contain any $^{14}C$ at all the compound-specific radiocarbon value is similar to the bulk. Therefore, a radiocarbon-modern sample, i.e., the topsoil composite, and the radiocarbon-dead lignite were prepared and measured repeatedly with different concentrations. The best fit for $F^{14}C_c$ and $m_c$
to match the observed $F^{14}C_m$ for both sets of measurements is calculated according to Haghipour et al. (2018).

The blank assessment (Figure 2) yields a contamination of $2.62 \pm 0.79\mu g$ C with a fraction modern of $0.59 \pm 0.18$, which is in range of previously determined contamination introduced by HPLC separation of lipids (e.g., Shah and Pearson, 2007; Birkholz et al., 2013). The impact of the constant contamination decreases as the sample mass increases. Therefore, the limit
towards large carbon masses of the fitted curve is equivalent to the radiocarbon signal of the sample unaffected by extraneously introduced carbon. For both samples, this limit and hence the compound $F^{14}C$ differs from the bulk $F^{14}C$ of the initial material. In the topsoil reference the compounds are depleted in radiocarbon ($F^{14}C$ = 0.94) with respect to the source, in the lignite the GDGTs are enriched ($F^{14}C$ = 0.06).

The recommended sample size to reach a precision <5% varies depending on the age of the sample. For samples with a
radiocarbon age < 1800 years ($F^{14}C$ > 0.8) a size of 20 $\mu g$ C is sufficient to reach the desired precision, while samples older than 6000 years ($F^{14}C$ < 0.5) require at least 50 $\mu g$ C. These uncertainties are taken into account when considering the GDGT $^{14}C$ results for the soil samples measured in this study.

### 3.2 Vertical Distributions of GDGTs

In the pre-alpine soil from Beatenberg, concentrations of GDGTs are generally highest in the topsoil, where the isoprenoid
and branched GDGTs are 10 and 38 $\mu g$ gdw$^{-1}$ respectively, whereas corresponding concentrations in the top soil layer of the Lausanne soil are much lower, 0.6 and 2 $\mu g$ gdw$^{-1}$, respectively (Figure 3). The concentration of both groups of GDGTs decreases sharply with increasing soil depths, with approximately ten times the abundance of isoGDGTs and brGDGTs in the top 5 cm than a few centimeters below. In contrast, isoprenoid and branched GDGTs concentrations normalized to organic carbon (OC) content increase with depth in the Beatenberg soil from $47\mu g$ gOC$^{-1}$ for isoGDGTs and from 175 $\mu g$ gOC$^{-1}$ for
brGDGTs in the top 5 cm to 273 $\mu g$ gOC$^{-1}$ and 80 $\mu g$ gOC$^{-1}$, respectively, between 20 and 40 cm depth. In the Lausanne soil profile, the OC-normalized isoprenoid and branched concentrations drop from 10 $\mu g$ gOC$^{-1}$ and 39 $\mu g$ gOC$^{-1}$, respectively, in the top 5 cm to 4 and 13 $\mu g$ gOC$^{-1}$ between 10 and 20 cm depth, and then increase to 14 and 12 $\mu g$ gOC$^{-1}$ between 60 and





80 cm.

The relative abundance of the individual brGDGTs also changes with soil depth, as reflected in the MBT'$_{5Me}$ and CBT' ratio
(De Jonge et al., 2014). The MBT'$_{5Me}$ index does not exhibit significant variability in either soil profile, while the CBT' index
increases with soil depth, especially in the Lausanne soil indicating a shift towards 6-methylated GDGTs (Figure 3).

### 3.3   Radiocarbon variations

The GDGT fractions prepared for AMS measurement contained between 30 and 80 μg C, except for the brGDGTs in the 10
to 20 cm depth interval in Beatenberg and the iso- and brGDGTs from 60 to 80 cm the Lausanne soil, which range between
15 and 20 μg C. The results of the radiocarbon measurements are shown in Figure 4, together with previously reported $^{14}$C
data for other soil carbon constituents (Van der Voort et al., 2017) . In the Beatenberg profile, radiocarbon signatures of both
isoprenoid and branched GDGTs closely follow the bulk OC with $\Delta^{14}C$ values of -23 and -30 ‰, respectively, in the top 5 cm,
and decreasing to -241 and -196 ‰ in the 20 to 40 cm depth interval, respectively. This contrasts with dissolved organic carbon
(DOC), which exhibits $^{14}$C-enriched and relatively invariant $\Delta^{14}C$ values throughout the soil profile (73 ‰ in the topsoil to
24 ‰ in the deeper soil)(Figure 4). Overall, $\Delta^{14}C$ values of both groups of GDGTs are similar to those of a $C_{29}$ n-alkane and
the long-chained (>$C_{26}$) n-alkanoic fatty acids (FA), but differ sharply from those of shorter-chained ($C_{16}$-$C_{22}$ measured on
the same samples (Van der Voort et al., 2017). The latter never reach $\Delta^{14}C$ values lower than -90 ‰ in the soils, resulting in
an offset between short-chained fatty acids and the other analyzed compounds that show stronger decreases with soil depth.
Similar patterns exist in the Lausanne profile: The $\Delta^{14}C$ values in isoprenoid and branched GDGTs decrease with depth from
-20 and -7‰ at 0 to 5 cm to -441 and -310 ‰ at 60 to 80 cm, respectively. While $\Delta^{14}C$ values of bulk OC and GDGTs parallel
one another closely in the Beatenberg soil, GDGT $\Delta^{14}C$ values are systematically lower than bulk OC and at each depth
interval in the Lausanne profile, with the difference between GDGTs and bulk OC ranging from -105 to -200 ‰. The GDGT
$\Delta^{14}C$ values are also more depleted than those of DOC and short-chain ($C_{16-22}$) FA in the soil, but are bracketed by n-$C_{27}$ n-
alkane and n-$C_{29}$ fatty acid $\Delta^{14}C$ values in the deepest soil section (Van der Voort et al., 2017). The $\Delta^{14}C$ values of the density
fractions from the same soil samples were also measured by Van der Voort et al. (2017). The low density fraction corresponds
to the free particulate organic carbon (free POC) and the high density fraction is interpreted as mineral-associated POC. Both
fractions do not differ by more than 40 ‰ in the top 20 cm of either soil profile, but in the lowest depth interval the fractions
diverge, with markedly lower $\Delta^{14}C$ values the for high density fraction, and values similar to DOC for the free POC fraction.
In both soils, the iso and brGDGTs exhibit similar or lower $\Delta^{14}C$ values than the high density fraction, mineral-associated
organic matter fraction.

### 3.4   Radiocarbon derived turnover times of GDGTs

Turnover times of the compounds are calculated based on a two-pool model that requires three parameters to be fitted: the
turnover time of the fast-cycling pool, the turnover time of the passive pool and the proportion of the fast-cycling pool. As only
one radiocarbon measurement per compound and depth interval is available, two of the parameters need to be estimated, while
one can be fitted accordingly. We use the proportion of the labile low-density fraction of the samples (Van der Voort et al.,





2017) to constrain the size of the fast-cycling pool. The turnover time of the fast-cycling pool can be estimated accordingly as the single-pool turnover time of the light fraction. Alternatively, the GDGT turnover in topsoil based on stable carbon isotopes has been shown to be similar to short-chain fatty acids (Weijers et al., 2010; Huguet et al., 2017). Thus, the turnover time of these compounds based on a single-pool box model can also be used to constrain turnover time of the fast-cycling pool of

GDGTs. For simplicity, a lag-term addressing the time between atmospheric carbon fixation and input into the soil is not used, as it is shorter than a decade (Solly et al., 2018) and its potential influence is hence already covered by the range of the turnover time estimates of the fast pool. The low $\Delta^{14}$C values of GDGTs in the deeper soil intervals result in a turnover time of the passive GDGT pool in the order of 1400 to 2000 years between 10 and 20 cm depth and 2000 up to 6000 years in the lowest depth interval in either soil (Table 1). These results are insensitive to changes of either the turnover time and the size of the

labile pool: A change of $\pm 10$ % of the proportion of the fast pool or $\pm 500$ years of the turnover time of the fast pool results in a maximum change of 10 % of the overall GDGT turnover time in the two lower depth intervals in either soil. Thus, despite the uncertainties in the estimation of the proportion of the pools and the turnover time of the fast pool, the turnover time of both groups of GDGTs clearly exceeds a millenium.

## 4 Discussion

### 4.1 Efficacy of GDGT isolation and $^{14}$C measurement protocol

Compared to prior methods to achieve individual isoGDGT separation by HPLC (Smittenberg et al., 2002; Ingalls et al., 2006), the introduced method isolates GDGTs only at the compound-class level, hence potential radiocarbon variations among GDGT isomers are not discernable. However, previous analyses of stable carbon isotopic as well as radiocarbon analysis of GDGTs on a molecular level do not show significant differences between the individual isoprenoid or branched GDGTs, respectively

(e.g., Ingalls et al., 2006; Shah et al., 2008; Oppermann et al., 2010; Weber et al., 2015). This implies similar metabolisms for brGDGT-producing organisms and also for microbial communities that synthesize isoGDGTs. Consequently, pooling of isomers within a compound class according to their respective microbial domain (bacteria, archaea) seems reasonable, particularly given the practical constraints imposed by their low abundance in many terrestrial (and aquatic) environments. The introduced method requires only a single normal-phase isolation step using the same columns that are used for quantification of

GDGTs (Hopmans et al., 2016), minimizing the time required for sample preparation and without extensive adjustments to the analytical HPLC set-up. The calculated contamination is in range of the blank assessment by Ingalls et al. (2006), but higher than the extraneous carbon observed by Birkholz et al. (2013). However, the blank assessment in Birkholz et al. (2013) is based only on a modern non-GDGT standard (cholesterol), potentially leading to an underestimation of the sample preparation blank.

The GDGT-specific $\Delta^{14}C$ values of the top soil and lignite samples used as "modern" and "fossil" endmembers for blank assessment did not yield values that fully matched those expected given their age. In case of the soil, different $\Delta^{14}C$ values of the GDGTs compared to bulk OC are to be expected due to the heterogeneous nature of soil organic matter, however for lignite sample that is of geologic age (> 30 Ma), all components would be expected to be radiocarbon-dead. A preliminary batch of





lignite that was extracted yielded 18 $\mu$g C of isoGDGTs and 48 $\mu$g C of brGDGTs, with corresponding $\Delta^{14}C$ values of the

resulting isolated compounds of of -960‰ and -980‰, respectively. The second batch of lignite used to assess constant contamination was prepared 4 months later and shows $\Delta^{14}C$ values consistently higher than -950‰ (figure 2). This shift towards higher $\Delta^{14}C$ values likely reflects contamination resulting from sample-to-sample carry-over on the HPLC. Although this is adressed in the blank assessment, this highlights the importance of repeated blank assessment in order to control for variations in carry-over and other potential sources of contamination (e.g., column bleed) over time. Careful assessment of compound

purity is also important to ensure robust isotopic determination.

## 4.2 Radiocarbon constraints on the origin and turnover of GDGTs in soils

Our study reveals low $\Delta^{14}C$ values, with corresponding radiocarbon ages of up to 6000 years for GDGTs in forested soils. These $^{14}$C characteristics are similar to those of the mineral-associated OM (from density fractionation), as well as long-chain, higher plant wax-derived n-fatty acids and n-alkanes. As GDGTs are microbial membrane lipids, these findings reveal the

presence of $^{14}$C-depleted, millennial age microbial residues as a component of organic matter in deeper soils. There are two possible pathways leading to these old apparent radiocarbon ages: (1) active GDGT-producing heterotrophic soil microbial communities in deeper soils are utilizing pre-aged SOM as a carbon source, and accrue this signal with continuous community turnover. Alternatively, (2) upon cell death these microbial lipids are stabilized for millenia, likely via interaction with soil minerals. We first consider the first explanation:

The $\Delta^{14}C$ values of living organisms, and their constituent lipids, directly reflect that of their metabolic carbon source as, unlike stable isotopes, they are impervious to biological fractionation effects (Ingalls and Pearson, 2005). Upon death of the organism, radioactive decay leads to depletion in $^{14}$C contents. Consequently, the $^{14}$C contents of iso- and brGDGTs should reflect that of the carbon source of their biological precursors. IsoGDGTs are known to be produced by Thaumarcheota and Euryarcheota (Schouten et al., 2013). The specific microbes that produce brGDGTs are yet to be identified, there is

strong evidence that the precursor organisms are heterotrophic bacteria (Pancost and Damsté, 2003; Weijers et al., 2010), with Acidobacteria amongst the candidate phyla (Damsté et al., 2018). For heterotrophic bacteria, potential carbon sources include DOC leached from the organic layer, exudates from root systems or organic matter that has accumulated during soil development. The activity of soil microbial communities has often been assayed using phospholipid-fatty acids (PLFAs), as phospholipids are only found in living cells and thus serve as biomarkers for viable microbial communities (e.g., Tunlid and

White, 1991). Compound-specific radiocarbon analyses of PLFAs have shown that soil microbes can use a variety of carbon sources, including "older" SOM (Kramer and Gleixner, 2006). However, root-derived C seems a dominant food source of heterotrophic microbial communities in temperate deciduous forest soils (Kramer et al., 2010) and this has been inferred to be a likely substrate for the producers of brGDGTs (Huguet et al., 2013). The turnover time of root carbon is on the order of decades at most (Gill and Jackson, 2000; Gaudinski et al., 2001; Solly et al., 2018)), and thus the old ages and long

turnover times of GDGTs observed in both soils analyzed in this study cannot be explained by the uptake of root-derived C. Accessible, labile carbon pools in the investigated soil profiles are represented by DOC and the light density fraction. These appear to be preferably used by microbial communities as evidenced by the $^{14}$C-enriched values of short-chain ($<C_{24}$) fatty





acids that likely reflect active microbial communities (Figure 4, 5). The markedly lower $\Delta^{14}C$ values of both isoGDGTs and brGDGTs at depth would require that both groups of precursor organisms, i.e., Archaea and Bacteria, occupy specific

niches using metabolic strategies that enable them to utilize stabilized, aged carbon. The precursor organisms of isoGDGTs in soils are known to be mainly comprised of crenarchaeota, i.e., chemoautotrophic nitrifiers using soil $CO_2$ as substrate (Leininger et al., 2006; Urich et al., 2008; Weijers et al., 2010; Damsté et al., 2012) and acetotrophic methanogens (Weijers et al., 2010). Contributions from the latter organisms in the studies soils are likely minor as the soils are not strictly anaerobic (Walthert, 2003)). This is also supported by GDGT-0/Crenarchaeol ratios, that differ sharply from those in soils and sediments

dominated by methanogens (Blaga et al., 2009; Weijers et al., 2010; Naeher et al., 2014). Soil-respired $CO_2$ has relatively high $\Delta^{14}C$ values (Gaudinski et al., 2000; Liu et al., 2006), and thus it seems highly unlikely that $^{14}$C-depleted signatures of isoGDGTs in the deeper soils results from metabolism of an old C substrate by active soil microbial communities. By analogy, the $^{14}$C-depleted characteristics of brGDGTs is difficult to reconcile with heterotrophic consumption of pre-aged C. Overall, the contrasting metabolisms of the GDGT precursor organisms (primarily autotrophy for isoGDGTs and heterotrophy

for brGDGTs), yet similar (and low) $\Delta^{14}C$ values for both compound classes, argue against an origin of the GDGT signals from microbial growth at depth.We therefore conclude that uptake of pre-aged carbon by active soil microbial communities is unlikely to be the cause for the $^{14}$C-depleted GDGT signatures.

We next consider the long-term stabilization microbially-derived carbon as the source of $^{14}$C-depleted GDGT signatures. This implies that microbial residues persist in soils for millennia, lending support to emerging concepts that microbial necro-

mass comprises an important component of older SOM (e.g., Lehmann and Kleber, 2015; Liang et al., 2017). Long-term persistence of GDGTs could arise from their stabilization by soil minerals at greater soil depths. The amphiphilic nature of lipids such as GDGTs, with both polar and hydrophobic components, promotes the association with mineral surfaces, and therefore may afford physical protection from degradation (Jandl et al., 2004; Kleber et al., 2007; von Lützow et al., 2008; Van der Voort et al., 2017). By comparison, in surface soils with high organic matter contents and less availability of reactive

mineral surfaces, GDGTs are continuously produced and degraded, which results in a younger mean radiocarbon age and evidence for turnover on decadal timescales (Weijers et al., 2010). This explanation agrees with conceptual models of soil organic matter dynamics whereby older SOM in deeper soils primarily consists of microbial metabolites that are stabilized by their interaction with mineral surfaces (Schmidt et al., 2011; Lehmann and Kleber, 2015). Given the structural resemblance between brGDGTs and isoGDGTs, and hence similar propensity to associate with mineral surfaces , we consider this a more likely

explanation for their similarly old $^{14}$C ages than "niche metabolisms" of different precursor organisms. The older GDGT age in the the Cambisol at Lausanne compared to the subalpine Podzol at Beatenberg with a bleached eluvial horizon also supports this conclusion. The higher contents of clay and highly reactive amorphous Fe and Al-oxides and hydroxides of the former (Table 2) are known to play a key role in the sorptive stabilization of SOM (Kaiser and Guggenberger, 2003; Kleber et al., 2007)

Overall, $^{14}$C characteristics of iso and brGDGTs and the inferred turnover times that are far longer than those of discrete POM (free light density fraction) and signature lipids of active microbial communities (short-chained fatty acids), but similar to those





of plant-derived long-chain n-alkanes and fatty acids (Figure 5), serve as strong evidence for the presence of mineral-stabilized microbial necromass in the studied forested mineral soils.

### 4.3 Implications for application of GDGTs as molecular proxies and soil tracer biomolecules

In addition to the insights into soil carbon turnover, the observed $^{14}$C signatures of GDGTs in the two soil profiles carry implications for their application as proxies of environmental conditions and as tracers of soil carbon input to aquatic environments. Several studies have shown that soils comprise a significant, and often the dominant, component of terrestrial organic carbon exported in the suspended load of rivers to lake and ocean sediments (e.g., Tao et al., 2015; Vonk et al., 2019; Hein et al., 2020). Prior analyses of branched GDGTs in sedimentary archives have revealed older GDGT ages than depositional ages (Smitten-

berg et al., 2005; Birkholz et al., 2013) that may reflect intermittent storage during transport or export of deeper mineral soil carbon suggesting a lag between production and deposition. Our findings suggest that this may be a consequence of protracted storage in and mobilization from deeper mineral soils. This reinforces the value of brGDGTs as tracers of soil carbon, but implies that much of the signal may be sourced from deeper soil layers, with corresponding GDGT proxy signals reflecting environmental conditions at the time that they were microbially produced.

### 325 5 Conclusions

We modified and validated a normal-phase HPLC method to isolate isoprenoid and branched GDGTs at the compound class level for radiocarbon analysis. Although further refinements in the method would be desirable, this new approach yields reliable GDGT $^{14}$C measurements on sample sizes > 20 $\mu$g C that have enabled novel questions to be addressed concerning the provenance and turnover of this key suite of microbial lipids. In addition to its application to questions of soil C cycling, the

streamlined method opens up new opportunities to further explore the biogeochemical and paleoclimate significance of this intriguing yet enigmatic class of lipids.

Application of the method to depth profiles for two well-studied sub-alpine soil profiles in Switzerland reveals a marked decrease in $^{14}$C contents of both isoGDGTs and brGDGTs with depth, with resulting model estimates for GDGT turnover times of 2000 to 6000 years in deeper mineral soils. These old ages for archaeal and bacterial membrane lipids provides

compelling evidence for stabilization of microbial necromass in soils that contributes to the long-term C storage. Through comparison with parallel $^{14}$C data for soil density fractions and other hydrophobic lipid biomarkers, we attribute the stability of GDGTs to protection via association with reactive mineral surfaces underlining the crucial role of microbial processes in soil C cycling and stabilization.

Our findings also provide motivation for further work to validate our interpretations and assess the broader significance of the

current limited suite of observations. For example, comparison of the proportions and isotopic signatures of intact polar lipid GDGTs relative to the "core" lipids measured here could shed light on the significance active GDGT-producing communities residing at a specific soil depth versus remnants of past microbial activity (necromass). Furthermore, while concentrations of isoprenoid as well as branched GDGTs commonly decrease with increasing soil depth (Huguet et al., 2010; Yamamoto et al.,

2016; Gocke et al., 2017), subsurface maxima in iso- and brGDGT concentrations have also been reported (Huguet et al.,
2010; Yamamoto et al., 2016)), potentially indicating depth-localized GDGT production. Future $^{14}$C analysis of GDGTs in
soil profiles that exhibit such sub-surface concentrations peak would be informative and provide context for our observations
in the two Swiss soil profiles. Further insights into the provenance and turnover of brGDGTs might be gained from in-depth
assessment of molecular distributions and associated proxy indices (De Jonge et al., 2014), that may reflect changes in current
or past microbial communities, or imply differences in susceptibility to degradation. Despite the presence of GDGTs as trace
constituents of SOM, their unequivocal microbial origin, distinctive chemical structures, and environmental properties that
their distributions encode render them powerful tracer compounds and molecular proxies. Here, we demonstrate that when
also constrained with natural abundance $^{14}$C , these compounds provide a new window into the role of microorganisms in soil
carbon cycling.

*Code and data availability.* The data set and script for the turnover model used in this study is available at https://doi.org/10.3929/ethz-b-
355 000430425

*Author contributions.* HG, ML and TE conceptualized the study, HG and DM designed the method, HG isolated the compounds and wrote
the turnover model, NH performed radiocarbon measurements, TSvdV provided data, HG, FH, ML and TE interpreted the data, HG prepared
the manuscript with contributions from all co-authors.

*Competing interests.* The authors declare that they have no competing interests

*Acknowledgements.* This study was funded by the Swiss National Science Foundation ("CAPS-LOCK2"&"CAPS-LOCK3"; #184865). We
thank Urs Overhoff and Markus Neuroth from the RWE Power AG for the provisioning of the lignite sample, and Marco Griepentrog
and Cindy de Jonge for helpful discussions. We thank the Laboratory for Ion Beam Physics (ETH) for support with the Accelerator Mass
Spectrometry measurements.



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

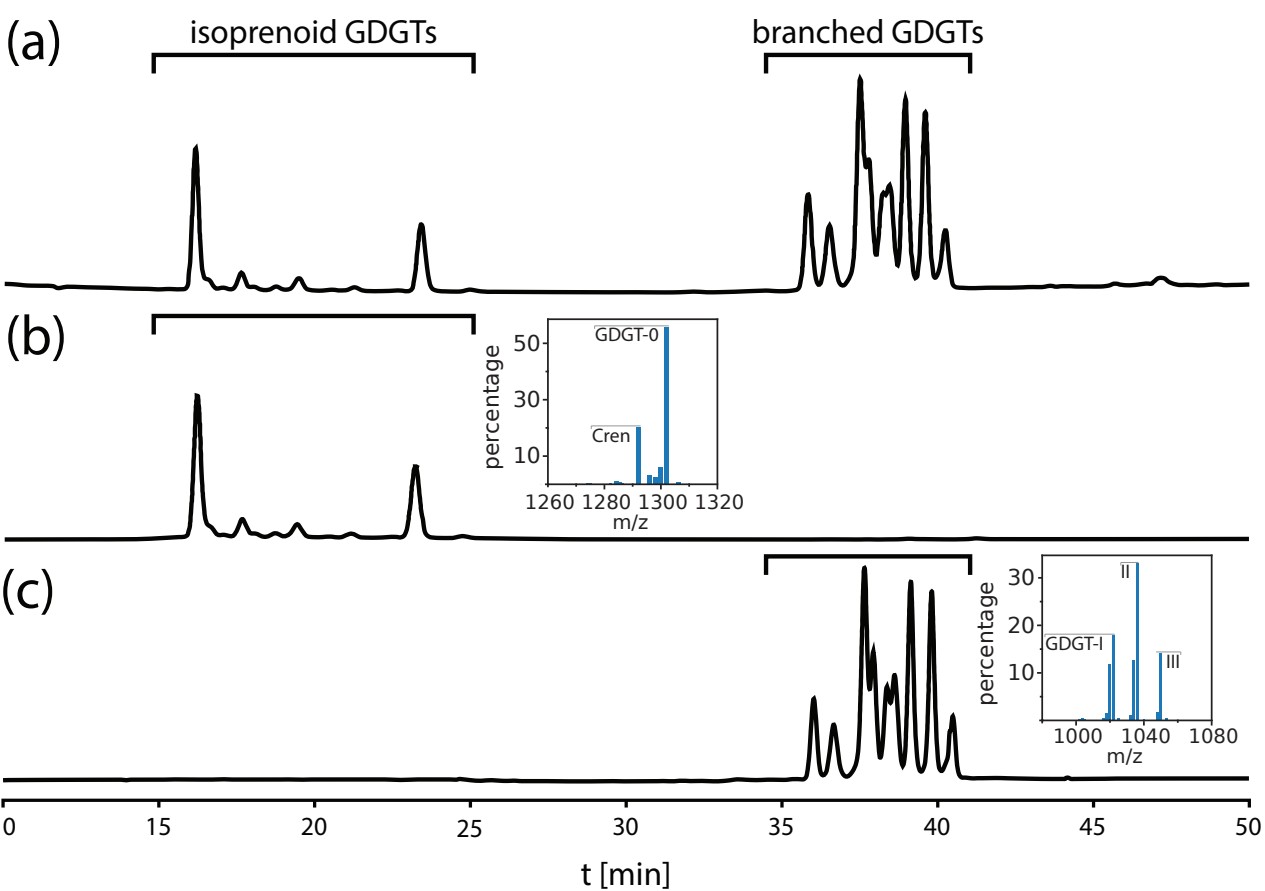

**Figure 1. HPLC mass chromatograms (*m/z* 500 - 1500) of GDGTs from a composition topsoil sample.** a) sample before GDGT isolation, b) the separated isoprenoid fraction and its composite mass spectrum (GDGT-3, GDGT-2 and GDGT-1 from left to right between Crenarcheol (Cren) and GDGT-0 are not labelled), c) the separated branched fraction and its composite mass spectrum (GDGT-I, II and III, corresponding to tetra-, penta- and hexamethylated GDGTs are labelled)(For molecular structures see Figure A1)





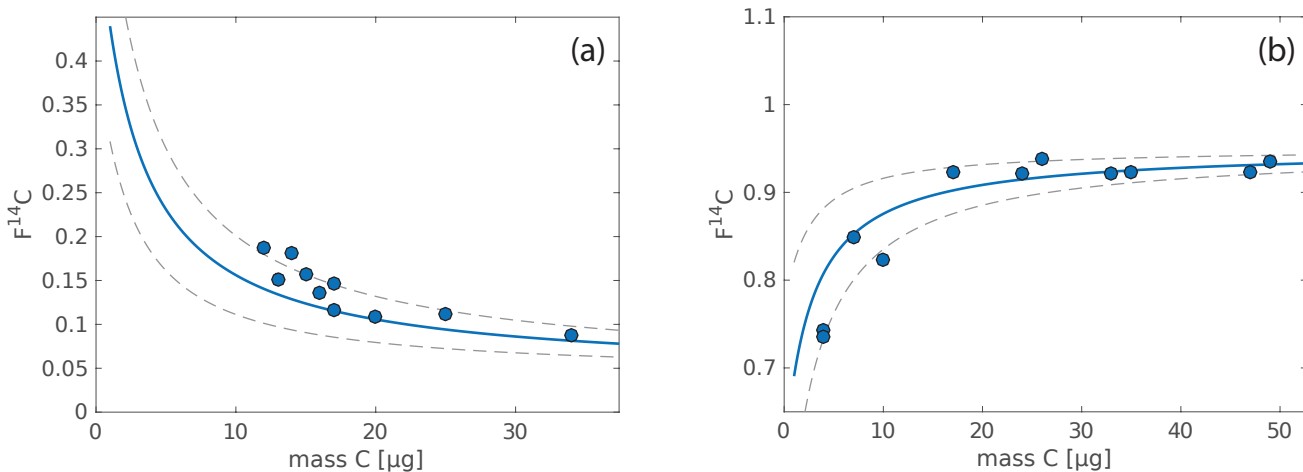

**Figure 2. Blank assessment associated with GDGT isolation.** a) The $F^{14}C$ dead standard (lignite) tracing a modern constant contamination of 1.55 $\mu$g C, b) The $F^{14}C$ modern standard (topsoil composite) tracing a radiocarbon dead constant contamination of 1.07 $\mu$g C. The solid line indicates the best fit of contamination mass and radiocarbon signal for both sets of samples considered jointly, the grey dotted lines show the $1\sigma$ error range

.



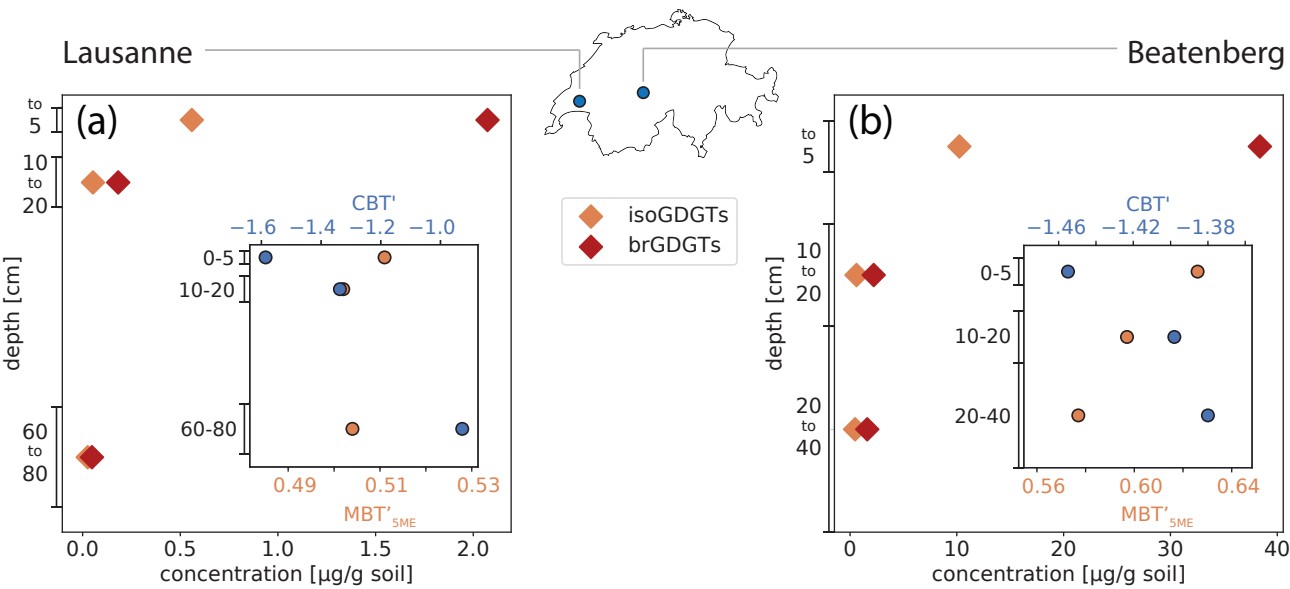

**Figure 3. BrGDGT abundances and parameter ratios in Lausanne and Beatenberg soil profiles.** In both soils (a - Lausanne, b - Beatenberg) the concentration of the samples per dry weight decreases rapidly with depth. The inner plots show the brGDGT-derived MBT'$_{5ME}$ (orange) and the CBT' (blue) indices in the respective soil. While the MBT'$_{5ME}$ does not vary a lot with depth, the CBT' increases significantly with depth in the Lausanne soil.

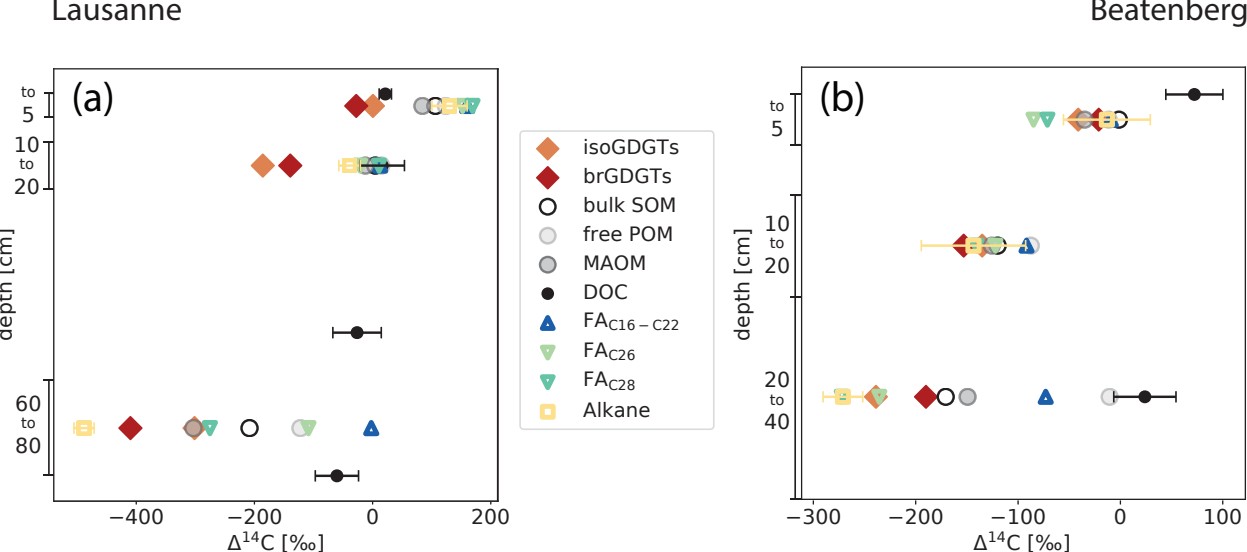

**Figure 4. Radiocarbon content, expressed as $\Delta^{14}C$ of organic matter fractions and compounds in Lausanne and Beatenberg soil profiles.** In both soils (Lausanne - A, Beatenberg - B) the radiocarbon contents of both branched and isoprenoid GDGTs decrease to a similar degree to that of bulk SOM in each profile. The measured n-alkane in the Beatenberg soil is $C_{29}$, in the Lausanne soil the n-$C_{27}$ homologue was analyzed. Alkane, n-fatty acid (FA), free particulate organic matter (POM), mineral-associated organic matter (MAOM) and dissolved organic carbon (DOC) $\Delta^{14}C$ values are taken from Van der Voort et al. (2017). DOC was not measured in the same intervals as the other parameters, but at 0, 15, 50 and 80 cm, and at 0 and 30 cm depth in the Lausanne and Beatenberg soils, respectively. If error bars are not visible, then the uncertainty is smaller than the symbol size.



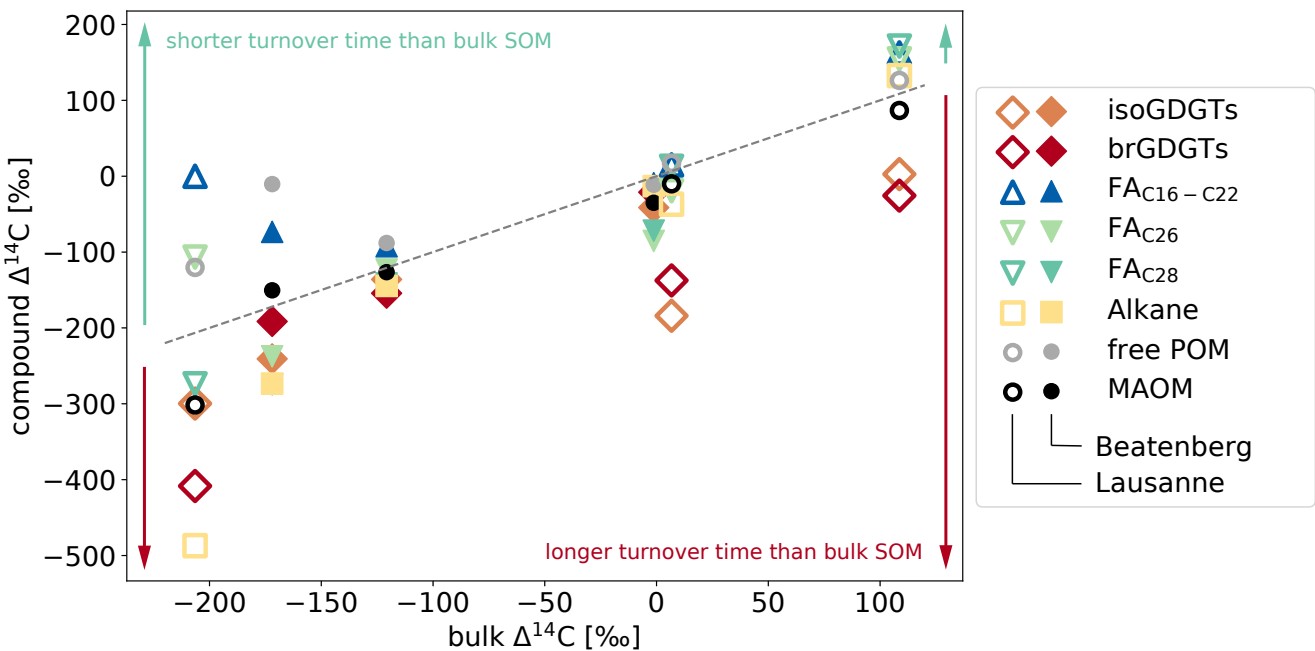

**Figure 5. Relationship between $\Delta^{14}C$ values of specific components and bulk SOM in Beatenberg and Lausanne soil profiles.** The different analyzed compounds and density fractions (Van der Voort et al., 2017) cover a greater range of $\Delta^{14}C$ values with soil depth as reflected in the ageing of the bulk OC in the respective soil. The dotted line represents equal compound and bulk $\Delta^{14}C$. Two groups are discernable: those with $\Delta^{14}C$ values higher than the bulk OC, and thus with a more rapid turnover (in most samples, this includes short-chain ($C_{16}$-$C_{22}$) FA and the low density fraction (free POM) and those with lower $\Delta^{14}C$ values implying longer turnover times (including long-chain n-alkanes (in the Beatenberg soil $C_{29}$-alkane, in the Lausanne soil the $C_{27}$), and fatty acids ($C_{26}$, $C_{28}$ FA) and the GDGTs.

**Table 1. Turnover times of isoprenoid and branched GDGTs.** Turnover times are based on the single-pool turnover time of the light fraction (fPOM) or the short-chain fatty acids (SCFA - in brackets) as approximation of the fast pool.

| loc | depth | fast pool proportion | turnover times fPOM (SCFA) [years] | | | | | |
|-----|-------|------|-----------|--------|--------|--------|--------|--------|
| | | | fast pool | | isoGDGT | | brGDGT | |
| Bb | 0-5cm | 0.897 | 350 | (340) | 710 | (740) | 440 | (450) |
| | 10-20cm | 0.193 | 890 | (920) | 1400 | (1400) | 1600 | (1600) |
| | 20-40cm | 0.115 | 350 | (760) | 2800 | (2700) | 2100 | (2000) |
| Ln | 0-5cm | 0.162 | 46 | (33) | 350 | (360) | 540 | (550) |
| | 10-20cm | 0.113 | 240 | (240) | 2000 | (2000) | 1400 | (1400) |
| | 60-80cm | 0.087 | 1200 | (300) | 3600 | (3700) | 5900 | (6100) |





**Table 2. Soil properties related to the stability of soil organic matter.** CEC (effective cation exchange capacity), $Fe_d$ (dithionite-extractable iron), $Fe_o$ and $Al_o$ (oxalate-extractable iron and aluminum), $Fe_p$ and $Al_p$ (pyrophosphate-extractable iron and aluminum) as well as sand, silt and clay content are provided by Zimmermann et al. (2006)

| loc | depth | CEC [mmolC kg$^{-1}$] | $Fe_d$ [ppm] | $Fe_o$ [ppm] | $Fe_p$ [ppm] | $Al_o$ [ppm] | $Al_p$ [ppm] | sand [%] | silt [%] | clay [%] |
|---|---|---|---|---|---|---|---|---|---|---|
| Bb | 0-5cm | 50 | 1047 | 684 | 505 | 538 | 571 | 84 | 7 | 8 |
| | 10-20cm | 15 | na | 133 | 99 | 280 | 268 | 83 | 15 | 3 |
| | 20-40cm | 37 | 5770 | 2183 | 2062 | 880 | 1713 | 80 | 14 | 6 |
| Ln | 0-5cm | 77 | 6187 | 3356 | 2900 | 1861 | 1304 | 62 | 25 | 13 |
| | 10-20cm | 61 | 6493 | 3039 | 2310 | 2030 | 1430 | 51 | 31 | 18 |
| | 60-80cm | 59 | 5840 | 2095 | 732 | 1156 | 791 | 57 | 27 | 16 |





isoprenoid GDGTs

| Name | | m/z |
|---|---|---|
| Crenarchaeol | | 1292 |
| Crenarchaeol' | | 1292' |
| GDGT - 0 | | 1302 |
| GDGT - 1 | | 1300 |
| GDGT - 2 | | 1298 |
| GDGT - 3 | | 1296 |

branched GDGTs

| Name | 6-methyl isomer | m/z |
|---|---|---|
| Ia | | 1022 |
| Ib | | 1020 |
| Ic | | 1018 |
| IIa / IIa' | | 1036 |
| IIb / IIb' | | 1034 |
| IIc / IIc' | | 1032 |
| IIIa / IIIa' | | 1050 |
| IIIb / IIIb' | | 1048 |
| IIIc / IIIc | | 1046 |

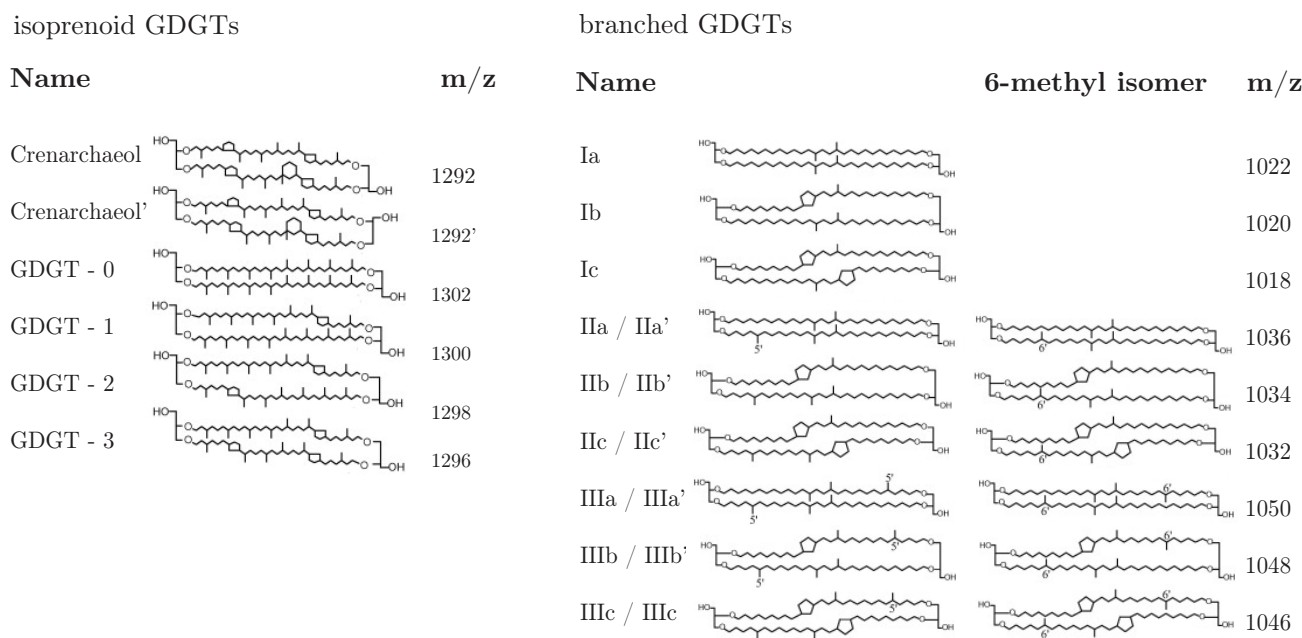

**Figure A1. Molecular structures of GDGTs analyzed in this study**