# Peer review of "Millennial-age GDGTs in forested mineral soils: 14C-based evidence for stabilization of microbial necromass"

_Biogeosciences, 2020_

## Referee Comment (RC1) · Anonymous Referee #1 · 20 Sep 2020

The manuscript titled "Millennial-age GDGTs in forested mineral soils: 14C-based evidence for stabilization of microbial necromass" by Gies et al., showcases an isolation method for both isoprenoid and branched glycerol dialkyl glycerol tetraethers (GDGTs) from mineral soils for subsequent compound-specific radiocarbon analysis. Based on the turnover times of millennia, the authors concluded a presence of stabilized microbial necromass in the soils. The manuscript encompasses a thorough description and validation of the methodological approach. It is well suited for publication for Biogeosciences. However, there are some errors and potential points for improvement. Firstly, this study uses newly produced GDGT data in comparison with previously published data of 2 sites in Switzerland. However the results from the individual sites are insuf-

ficiently discussed. For example, regarding Fig. 5, there is only one reference line for the turnover time for bulk SOM even though your study features 2 separate sites. Both GDGT and additional data from Van der Voort et al. (2007) partially differ between the sites. Consider showing individual reference lines for bulk SOM and discuss how the differences between data from both sites do or do not affect your overall conclusion. Secondly, there is no explanation of the MBT'5Me and the CBT' ratio. The results for the ratios are currently not discussed and do not contribute to the overall conclusion. This could be integrated better. Thirdly, there are some typing errors and additional suggestions, which are listed below.

Line 28: Since you already introduced 14C as a term for radiocarbon, you should keep to writing 14C instead of radiocarbon

Line 33: Starting from line 33, you inserted an indentation at the beginning of a paragraph. You should either dismiss the indentation or do it constant throughout the manuscript.

Line 38: "the relative abundance of branched versus isoprenoid GDGTs..." Since you already introduced the abbreviation brGDGTs and isoGDGTs, you should just write:" the relative abundance of brGDGTs versus isoGDGTs..."

Line 40: See line 38.

Line 44: "…and soil pH Weijers et al. (2007), respectively…" Weijers et al, 2007 in brackets.

Line 53: Apparently, this paragraph was falsely edited by the authors since punctuation is placed in the middle of a sentence. This should be re-written.

Line 63: Again, since you already introduced isoGDGTs and brGDGTs, you should stick to the abbreviations.

Chapter 2.1 Study site: You may want to consider incorporating a map showing the study sites. Or you may also refer to Fig. 3 to show where the study sites are located.

Line 94: "As extracting several kg of material is impractical, focused instead on pooled isolation and 14C measurement of isoprenoid GDGTs and branched GDGTs, respectively. . ." I assume, a "we" is missing between impractical and focused.

Line 97: "For this study, the pooling of the GDGTs reduced the required initial sample size to a maximum of 500 gdw of soil." How exactly was the pooling performed? What compounds were pooled together?

Line 100: "In brief:" no need for a new paragraph.

Line 112: C46 subscript.

Line 179: What is the "MBT'5Me? And the CBT' ratio?

Line 260: "The _14C values of living organisms, and their constituent lipids, directly reflect that of their metabolic carbon source as, unlike stable isotopes, they are impervious to biological fractionation effects (Ingalls and Pearson, 2005)." This is sentenced quite cumbersome. I would suggest rephrasing.

Line 344 & 346: You should stick to writing subsurface or sub-surface.

Line 346: ". . .that exhibit such sub-surface concentrations peak.." Either concentration peaks or exhibit such a sub-surface concentrations peak.

---

## Referee Comment (RC2) · Anonymous Referee #2 · 22 Sep 2020

Gies et al have developed a method to isolate and prepare GDGTs for radiocarbon analysis. In their study, they present 14C data for isoprenoid and branched GDGTs in two well studied soil profiles in Switzerland. They find that GDGTs have an increasingly longer turnover time with depth, and suggest that GDGTs are protected against degradation through associations with mineral surfaces. Their data provide evidence for the contribution of microbial necromass to the pool of stabilized carbon in (deep) soils.

The data and the paper are well suited to be finally published in Biogeosciences. However, I have some comments, questions, and suggestions.

[Figure]

General comments: I miss some basic background information in the introduction. Details on the sources/producers of GDGTs in soils, as well as their metabolism is crucial for this paper, but these aspects are only marginally addressed in the current version. It is not so clear on what level the 14C composition of iso and branched GDGTs will be interpreted: as markers for Archaea and Bacteria, or on a more specific level (e.g. Acidobacteria in case of the branched GDGTs)? The introduction should at least summarize what is known about the producing organism(s) of GDGTs. For the branched GDGTs, at least mention that Ia has been found in specific Acidobacteria, although it has not been excluded that other phyla may produce brGDGTs as well (Sinninghe Damsté et al., 2011, 2014, 2018). Also mention the (presumed) metabolism (for example Pancost and Sinninghe Damsté, 2003; Oppermann et al., 2010; Weijers et al., 2010). Add similar information for the isoGDGTs. They are merely introduced as Archaea, but this is of course a very broad term. Do they have the same producers as in the marine environment? What is their role in soils? Again Sinninghe Damsté (et al., 2012) has done some work on this. I found part of this missing information in the discussion (section 4.2), but I really think this should be part of the introduction already.

In addition, there are a couple of studies that provide estimates of the turnover of branched GDGTs in soils. Although they have not used 14C dating, these studies are currently not mentioned in the introduction. Check Weijers et al., 2010; Peterse et al., 2010; Huguet et al 2013, 2017, who propose turnover rates of years to decades for branched GDGTs in surface soils, each following a different approach.

Finally, the implications of the results for both soil organic carbon cycling as well as for paleoclimate reconstruction have not been given sufficient attention. The authors mention that there are implications for the use of GDGTs as 'tracers and proxies', but it is not clear what those will be exactly. MBT'5me and CBT' ratio values are reported in the results, but these indices are nowhere explained, and the values are not discussed. How will the turnover rates estimated in this study influence GDGT-based paleoclimate reconstructions? One of the few implications that is mentioned in

the implications section (4.3), is that old brGDGTs (older than the depositional age) in sedimentary archives reported in the literature confirm a soil source of brGDGTs, as the old age may be explained by intermittent storage during transport, or export of deeper soil material. This is actually in contrast with the overwhelming evidence for the production of brGDGTs in aquatic systems (rivers, lakes, and the coastal marine environment, see references in detailed comment below, as well as De Jonge et al., 2014 for in situ production in rivers and Sinninghe Damsté 2016 for in situ production in coastal settings), and the offsets in brGDGT signals between e.g. lake sediment and catchment soils (for example, see Tierney and Russell 2009; Miller et al., 2018; Guo et al., 2020). The aspect of in situ production and thus a contribution of aquatic sourced GDGTs (both branched and isoprenoid) has not been mentioned in the manuscript, and should be addressed in the implication section (and introduction, where appropriate).

Detailed and textual comments: P2L54-55: This sentence seems incomplete. P3L56: soils are considered a major source of brGDGTs to aquatic systems: I am not convinced that this is still widely believed. Over the past few years, several studies have provided evidence for a primarily in situ, hence aquatic source of brGDGTs in lakes (e.g. Tierney and Russell, 2009; Sinninghe Damsté et al., 2009; Tierney et al., 2010; Weber et al., 2015, 2018; Miller et al., 2018; Russell et al., 2018; van Bree et al., 2020). P3, section 2.1: also add the amount of samples included in this study. It later appears that one sample from every soil horizon has been analyzed. Provide this information here. P3L82-85: this sentence is very hard to read. It even seems like there may be some words missing? P4L94: . . .is impractical, WE focused instead on. . . P4L96: Adding to my earlier comment: the authors should better introduce the (supposed) sources of the GDGTs in soils in their introduction, and indicate here on what level their putative biological precursors and formation pathways are considered common. For brGDGTs, this may be true on the level of 'bacteria', but there are indications that the 5-methyl and 6-methyl isomers are produced by different subdivisions of the phylum Acidobacteria (Sinninghe Damsté et al., 2018). Given that the authors have pooled the 5-methyl and 6-methyl brGDGTs in this study, this is important information to add.

The same is true for isoGDGTs, of course. P4L124: Was the purity check done on full scan, or using selected ion monitoring as mentioned in Hopmans et al., 2016? In the latter case, how can you assure that there were no potentially co-eluting compounds present? P5 section 2.4: can you give a slightly more elaborate description of the different pools? What kind of compounds do you expect, or are assumed to be part of the fast cycling and passive pools, respectively? P6L164-168: Given the uncertainties associated with the sample size, and the minimal sample sizes mentioned here (at least 20 ugC for samples with a radiocarbon age <1800) and at least 50 ugC for samples older than 6000) compared to the relatively small sample size used in this study (at least 15 ugC), I think it is important to include the sample sizes of each sample in a (supplementary) data table. P7L179: The MBT'5me and CBT' ratios have not been defined. P7L199: Do you mean C28 fatty acid? P8L212: what do you mean with 'light fraction'? P8L212-213: Can you add the range of the estimated turnover times, for reference? How do the turnover times based on the two different approaches compare? P8 section 4.1: I think that the motivation to pool all GDGTs for radiocarbon measurements comes a bit late, as the pooling is a passed station. What if the literature had pointed out that each GDGT was likely to have another isotopic composition? How would you then interpret the pooled results? I feel like this section on the presumed shared metabolism should be moved to the introduction, as I am sure that this information has influenced the approach of this study rather than formed the 'coincidental happy fit' as presented now. P9 section 4.2: same comment here. The metabolism of the GDGTs, and its consequences for their 14C composition, should be better described in the introduction. P10L281: Thaumarchaeota L283: replace studies by studied L301: How do you match the previously reported turnover rates of GDGTs in surface soils of years to decades to centennia to even millennia, as you find here? L323: as mentioned earlier, I am not convinced that brGDGTs systematically trace soil OC given the evidence for in situ production in aquatic systems (rivers and lakes), or the loss of the soil signal upon entering a river (Zell et al., 2013, 2014; Guo et al., 2020). L335-337: Is there any way we can test the protection of GDGTs through association

with mineral surfaces?

General: Damsté et al should be Sinninghe Damsté et al.

Figures: The panels in the figures are almost consistently plotted the other way around than how they are discussed in the text. For example, the Beatenberg profile is described first in section 2.1, but GDGT concentrations are given in panel B of Figure 3. Similarly, results from the Beatenberg profile are presented first in the text (3.3), but are visualized in panel B of figure 4. Can the panels in the figures be turned around to avoid confusion?

References: van Bree et al., 2020 Biogeosciences Discussions De Jonge et al., 2014 GCA – Yenisei river Guo et al., 2020 Biogeosciences Hopmans et al., 2016 Org Geochem. Huguet et al., 2013 GCA 105, 294-315 Huguet et al., 2017 GCA 203 103-116 Miller et al., 2018 Climate of the Past Oppermann et al., 2010 GCA Pancost and Sinninghe Damsté, 2003 Chem Geol. Peterse et al., 2010 OG Russell et al., Org Geochem Sinninghe Damsté et al 2009 GCA Sinninghe Damsté et al 2011 AEM Sinninghe Damsté et al 2012 AEM Sinninghe Damsté et al 2014 AEM Sinninghe Damste, 2016 GCA Sinninghe Damsté et al 2018 Org Geochem Tierney and Russell, 2009 OG Tierney et al., 2010 GCA Weber et al, 2015 GCA Weber et al, 2018 PNAS Weijers et al., 2010 Biogeosciences Zell et al 2013 L&O Zell et al 2014 Biogeosciences

---

## Author Comment (AC1) · 21 Oct 2020

We thank Referee #1 for their constructive comments and have implemented them to improve our manuscript. The Referee points out that they would like more discussion of potential causes for the differences in compound-specific radiocarbon values between the two sites and that the implications of the MBT'$_{5ME}$ and CBT' indices are not sufficiently discussed. We agree that the referee's remarks are justified and addressed their suggestions as follows:

**Referee: The results from the individual sites are insufficiently discussed. For example, regarding Fig. 5, there is only one reference line for the turnover time**

[Figure]

**for bulk SOM even though your study features 2 separate sites. Both GDGT and additional data from Van der Voort et al. (2007) partially differ between the sites. Consider showing individual reference lines for bulk SOM and discuss how the differences between data from both sites do or do not affect your overall conclusion.**

**Authors' response:** The dotted line is the 1:1 line for bulk $^{14}$C on the x-axis and compound $^{14}$C on the y-axis, thus the line is the same for both sites, which are distinguishable by the filled and open symbols. A "1:1" label was added to Figure 5 to avoid confusion. We agree that the discussion would benefit from elaborating on the differences between both sites, and have hence modified the paragraph at the end of section 4.2:

*"[. . .] The older GDGT 14C age in the lowest depth interval of the Cambisol at Lausanne compared to the subalpine Podzol at Beatenberg with a bleached eluvial horizon also supports this conclusion. The Lausanne soil has higher contents of clay and highly reactive amorphous Fe and Al-oxides and hydroxides (Table 2) which are known to play a key role in the sorptive stabilization of SOM (Kaiser and Guggenberger 2003; Kleber et al., 2007). The 14C signatures of GDGTs are similar in the top 20 cm at both locations (Figure 4), however, in the Lausanne soil, the alkanes and fatty acids are less depleted in $^{14}$C compared to Beatenberg resulting in an offset between GDGTs and the plant-derived compounds. One explanation could be the different thicknesses of the organic layer at the two sites (Walthert et al., 2003.) The 20 cm thick organic layer at Beatenberg retards the inputs of plant-derived C into the mineral soil and thus leads to longer turnover times of bulk OC in the topsoil compared to Lausanne, where the organic layer is only 2 to 3 cm thick. Contrary to the plant wax components, the turnover of isoGDGTs and brGDGTs does not seem to be affected by the thickness of the organic layer, resulting in the greater age offset observed in the Lausanne soil."*

**Referee: There is no explanation of the MBT'$_{5ME}$ and the CBT' ratio. The results for the ratios are currently not discussed and do not contribute to the overall**

**conclusion.**

**Authors' response:** The primary focus of this manuscript is on the use of GDGTs as tracers of soil microbial bio-/necro-mass, rather than the proxy information residing in their molecular distributions. Nevertheless, we agree that the implications of the MBT'5ME and CBT' ratio should be discussed in more detail. The equations how to calculate the indices are added to Figure A1, additionally we have expanded the paragraph at the end of section 3.2:

*"Changes in the relative abundance of the individual brGDGTs (Figure A1) are reflected in the Methylation of Branched Tetraethers (MBT'$_{5ME}$) index and in the Cyclisation of Branched Tetraethers (CBT') index, with these parameters increasing with higher proportions of methyl groups and cyclopentane moieties, respectively, in the brGDGT structures (de Jonge et al. 2014). In the Lausanne soil, the MBT'$_{5ME}$ is largely invariant with depth, while the CBT' increases from -1.59 to -0.93 (Figure 3), indicating a higher proportion of GDGTs with cyclopentane moieties in the subsoil. In the Beatenberg soil, a more modest increase of the CBT' index is evident, with a change from -1.46 to -1.38, while the MBT'$_{5ME}$ decreases slightly from 0.63 to 0.58."*

The implications of these results are further discussed in a paragraph in section 4.3:

*"The relative abundance of different brGDGTs as expressed in the MBT'$_{5ME}$ and CBT' index values correlates with MAT and soil pH (Weijers et al., 2007). The changing proportions of the individual brGDGTs reflected in both soils in the increasing CBT' index and, in the Beatenberg soil, decreasing MBT'$_{5ME}$ index with depth correspond to a pH change from 4.6 to 5.7 in Lausanne and 4.8 to 5 in Beatenberg, while in the latter the reconstructed MAT decreases from 10.9°C to 9.5°C. In Beatenberg, these reconstructed values do not match the measured pH change from 3.7 to 4.4 with depth, or reflect the MAT of 4.6°C. Nevertheless, the direction of the changes, the increase of pH and the decrease of temperature with depth, is echoed in the relative abundance of GDGTs. In Lausanne, however, the brGDGT-based increase in pH is not observed*

*in the soil values, with measured pH remaining largely invariant (4.6 to 4.5) throughout the profile. Therefore, the significant increase of the CBT' index and hence the higher relative proportion of brGDGTs with cyclopentane moieties might reflect preferential association of these compounds to mineral surfaces compared to those without cyclisation. Further work is needed to ascertain whether some GDGT structures are more prone to protection by mineral association than others, as a change in relative abundance of brGDGTs with time due to different turnover of individual GDGTs would need to be considered when using brGDGTs to reconstruct environmental conditions."*

**Additional minor suggestions from the Referee:**

**Line 97: How exactly was the pooling performed? What compounds were pooled together?**

**Authors' response:** Instead of separating individual molecules, e.g., GDGT-0, GDGT-1, GDGT-2 etc. (see figure A1), we used a single time window for the collection of isoGDGTs and brGDGTs respectively. In this way, the molecules are pooled on the compound class level.

line 95-97 was changed to *"As sampling and extracting several kg of material is impractical, we did not attempt to isolate individual GDGT compounds (Figure A1), but focused instead on pooled compound class-level isolation and $^{14}C$ measurement of isoprenoid GDGTs and of branched GDGTs, respectively, based on the premise that there are common putative biological percursors and biosynthetic formation pathways for each compound class (Schouten et al., 2013)"*

**All typing errors, formatting suggestions and bulky sentences highlighted by the referee have been addressed.**

---

## Author Comment (AC2) · 21 Oct 2020

We thank Referee #2 for their thorough and helpful comments and have edited our manuscript accordingly. The Referee raises the issue that GDGTs in general, their turnover and especially the metabolism of their putative precursors are not sufficiently introduced, especially considering that our method setup and conclusions are based on these assumptions. We agree with this assessment and have hence expanded the introduction to improve the manuscript. Additionally, Referee #2 points out that the implications for the use of GDGTs as proxies and tracers have not been sufficiently discussed. We have expanded discussion in the corresponding paragraph to also ad-

dress the production of GDGTs in aquatic environments. However, as the stability of microbial remnants in soils is the primary focus of this study, we address possible implications as motivation for further research.

**Referee: I miss some basic background information in the introduction. Details on the sources/producers of GDGTs in soils, as well as their metabolism is crucial for this paper, but these aspects are only marginally addressed in the current version. [...] In addition, there are a couple of studies that provide estimates of the turnover of branched GDGTs in soils. Although they have not used [14]C dating, these studies are currently not mentioned in the introduction.**

**P3 L56: soils are considered a major source of brGDGTs to aquatic systems: I am not convinced that this is still widely believed. Over the past few years, several studies have provided evidence for a primarily in situ, hence aquatic source of brGDGTs in lakes.**

**P4 L96: Adding to my earlier comment: the authors should better introduce the (supposed) sources of the GDGTs in soils in their introduction, and indicate here on what level their putative biological precursors and formation pathways are considered common. For brGDGTs, this may be true on the level of 'bacteria', but there are indications that the 5-methyl and 6-methyl isomers are produced by different subdivisions of the phylum Acidobacteria (Sinninghe Damsté et al., 2018). Given that the authors have pooled the 5-methyl and 6-methyl brGDGTs in this study, this is important information to add.**

**P8 section 4.1: I think that the motivation to pool all GDGTs for radiocarbon measurements comes a bit late, as the pooling is a passed station. What if the literature had pointed out that each GDGT was likely to have another isotopic composition? How would you then interpret the pooled results? I feel like this section on the presumed shared metabolism should be moved to the introduction.**

**Authors' response:** We reworked the introduction to include more of the information suggested by the referee. Some of this information was previously embedded in the Discussion, but we recognize that it would be beneficial to include more background in the introductory section:

[...]

[revised manuscript text omitted]

**Referee: Finally, the implications of the results for both soil organic carbon cycling as well as for paleoclimate reconstruction have not been given sufficient attention. The authors mention that there are implications for the use of GDGTs as 'tracers and proxies', but it is not clear what those will be exactly. MBT'5me and CBT' ratio values are reported in the results, but these indices are nowhere explained, and the values are not discussed. The aspect of in situ production and thus a contribution of aquatic sourced GDGTs (both branched and isoprenoid) has not been mentioned in the manuscript, and should be addressed in the implication section (and introduction, where appropriate).**

**P10 L323: as mentioned earlier, I am not convinced that brGDGTs systematically trace soil OC given the evidence for in situ production in aquatic systems (rivers and lakes), or the loss of the soil signal upon entering a river**

**Authors' response:** We agree with the Referee (and Referee #1 who raised the same issue) that we did not discuss or provide context for the MBT'$_{5ME}$ and CBT' indices sufficiently. Thus, we have added paragraphs in section 3.2 and 4.3 describing the variations of these indices within the cores and discussing the implications of these results, respectively (for the exact wording of these paragraphs see the Authors' response for Referee #1).

We have also added information regarding the aquatic production of GDGTs to the introduction and also reworked section 4.3. Nevertheless, we kept this section rather short, as the main focus of this manuscript is on GDGTs dynamics in soils and we are in the process of applying the GDGT separation method to river sediments in order to investigate the GDGT-specific radiocarbon signals in aquatic environments and how they compare to soils:

*"In addition to the insights into soil carbon turnover, the observed $^{14}C$ signatures of*

*GDGTs in the two soil profiles carry implications for their application as proxies of environmental conditions and as tracers of soil carbon input to aquatic environments. The putative application of brGDGTs as soil tracers has been undermined by the growing evidence pointing towards in-situ production as a major source of brGDGTs in aquatic environments (e.g., de Jonge et al., 2014; Sinninghe Damsté, 2016; Miller et al., 2018; Guo et al., 2020). Nevertheless, prior radiocarbon analyses of branched GDGTs in aquatic sediment sequences have revealed older GDGT ages than depositional ages (Smittenberg et al., 2005; Birkholz et al., 2013), consistent with a contribution of aged brGDGTs that were subjected to protracted storage in and mobilization from deeper mineral soils."*

**Further detailed and textual comments:**

**P3, section 2.1: also add the amount of samples included in this study. It later appears that one sample from every soil horizon has been analyzed. Provide this information here.**

We added this information as suggested:

*"At each site, soil cores were taken on 16 locations on a regular grid on a 1600 m² plot following protocols implemented as part of the LWF sampling program (van der Voort, 2016) and bulked to yield representative samples for three depth layers at each site: a sample from the A-Horizon comprising the top 5 cm of the soil , a second ranging from 10 to 20 cm depth, and a third from the B-Horizon between 20 to 40 cm and 60 to 80 cm depth at the Beatenberg and Lausanne site, respectively."*

**P4 L124: Was the purity check done on full scan, or using selected ion monitoring as mentioned in Hopmans et al., 2016? In the latter case, how can you assure that there were no potentially co-eluting compounds present?**

The aliquot of the polar fraction with the internal standard that was used to determine the concentration of GDGTs was measured using selected ion monitoring, while the

purity check on the isolated fractions for $^{14}$C analysis was performed in full scan mode. We added this information to line 124:

*"The isolated compound classes and the subset of the initial polar fraction set aside previously were analyzed for purity and quantification using the same HPLC system coupled to a quadrupole mass spectrometer (Agilent 6130) according to Hopmans et al. (2016) with the exception that the purity of the isolated fractions was tested in full scan instead of selected ion monitoring mode."*

**P5 section 2.4: can you give a slightly more elaborate description of the different pools? What kind of compounds do you expect, or are assumed to be part of the fast cycling and passive pools, respectively?**

The concept behind the two-pool model is that soil organic matter cycles on a continuum of time scales, even at the compound scale, as despite their same structure the compounds may vary in their degree of interaction with minerals or aggregates, impacting their stabilization and decomposition rate. Thus, the fast and the slow pool are a simplified representation of GDGTs with different decomposition rates.

**P6L164-168: Given the uncertainties associated with the sample size, and the minimal sample sizes mentioned here (at least 20 ugC for samples with a radiocarbon age <1800) and at least 50 ugC for samples older than 6000) compared to the relatively small sample size used in this study (at least 15 ugC), I think it is important to include the sample sizes of each sample in a (supplementary) data table.**

We agree, and have hence added the amount of C in the supplementary data table. The sample sizes mentioned here are not required minimum sample sizes, but the recommended sample size needed to keep the amount of the extraneously added carbon below 5% of the total measured carbon. Smaller samples are possible, but the resulting uncertainties will correspondingly be higher.

**P8L212: what do you mean with 'light fraction'?**

The 'light' fraction is the low-density fraction mentioned in the previous sentence. We will add the information that this fraction is *"the so-called particulate organic matter consisting of little decomposed residues"* and restructure the sentence to make this clearer. The preparation of this fraction is described in the referenced publication by Van der Voort et al. (2017): dried soil was immersed in sodiumpolytungstate at a density of 1.6 g/cm$^3$ in a 5:1 (v:v) ratio and first left to settle for 1 hour, then centrifuged at 500 rpm for 20 minutes. After centrifugation, the floating material was separated on a pre-combusted 0.7 mm glass fiber filter (GFF) and rinsed with deionized (DI) water.

**P8L212-213: Can you add the range of the estimated turnover times, for reference? How do the turnover times based on the two different approaches compare?**

We have added a reference to Table 1, where the turnover times of the fast pool are listed. The two different approaches lead to GDGT turnover times that differ by less than 5%; we have added this information to the text as follows:

*"[...] its potential influence is hence already covered by the range of the turnover time estimates of the fast pool (Table 1), the resulting GDGT turnover times based on either short-chain FAs or the light fraction differ by less than 5%."*

**P10 L301: How do you match the previously reported turnover rates of GDGTs in surface soils of years to decades to centennia to even millennia, as you find here?**

In the study by Weijers et al. (2010), the turnover times of brGDGTs are found to be similar to the turnover times of long-chain fatty acids and slightly shorter than long-chain n-alkanes. In the surface soil at the Beatenberg site, the turnover times of brGDGTs are in range with the turnover times of these other compounds, hence our findings of centennial turnover times do not contradict with these previously reported

turnover rates of decades considering that all studied compounds in this soil have turnover times on the order of hundreds of years. In the Lausanne soil, our calculation assumes that the fast cycling pool of GDGTs has a turnover time of a few decades similar to the short-chain fatty acids in this soil. However, the low $\Delta^{14}$C values of GDGTs require an additional slow cycling fraction of GDGTs that increases the overall turnover time to several hundred years. The methods used by Weijers et al. (2010) (single-pool model using stable carbon isotopes) and Huguet et al. (2017) (dividing lipid concentration in sample by incubation production rate) are not capturing the stabilized fraction of GDGTs with a slow turnover time as they trace the turnover of recent C inputs into the soil, yielding relative fast cycling rates (Trumbore, 2000). Here, we can only guess why GDGTs turn over especially slow in the Lausanne soil, but given that the the Lausanne topsoil had high contents of clay and highly reactive amorphous Fe and Al-oxides and hydroxides, it seems likely that stabilization by the interaction with mineral surfaces is a primary reason. This slow-cycling fraction is likely predominating in deeper soil horizons, manifesting itself through the observed old $^{14}$C ages (millennial turnover rates).

We address the reason for the slow GDGT turnover at the end of section 4.2:

*"[...] The older GDGT $^{14}$C age in the lowest depth interval of the Cambisol at Lausanne compared to the subalpine Podzol at Beatenberg with a bleached eluvial horizon also supports this conclusion. The Lausanne soil has higher contents of clay and highly reactive amorphous Fe and Al-oxides and hydroxides (Table 2) which are known to play a key role in the sorptive stabilization of SOM (Kaiser and Guggenberger 2003; Kleber et al., 2007)."*

**L335-337: Is there any way we can test the protection of GDGTs through association with mineral surfaces?**

A possibility would be to test whether mineral surface area and GDGT concentration correlate in a broad range of soil and sediment samples that differ in their organic matter and mineral content and composition. Also, density fractions of soils could be

analyzed for their concentration of GDGTs and their respective [14]C ages. An increase of GDGTs relative to organic carbon in the high-density (mineral-associated) fraction compared to the fractions of lower density would support the assumption that a significant proportion of GDGTs in soils are closely associated with mineral surfaces. An additional approach would be to measure GDGTS directly in mineral-protected OM following oxidization with sodium hypochlorite and subsequent dissolution of minerals by 10% hydrofluoric acid (HF) (Mikutta et al., 2006). We added a sentence in section 5:

*"The potential sorptive stabilization of GDGTs could be verified by measuring GDGTS and their 14C contents directly in mineral-protected OM (Mikutta et al., 2006), which, however, could be hindered by their low concentration.*

**All typing errors, formatting suggestions and grammar issues highlighted by the referee have been addressed. Also, the inverse order of figure panels compared to discussion in the text is curious and could indeed be confusing, but as the figure panels in Figure 3 are arranged according to the position of the samples in the map it was actually more convenient to just swap the respective paragraphs in the text to solve this issue.**